# Heme Oxygenase Protects against Placental Vascular Inflammation and Abortion by the Alarmin Heme in Mice

**DOI:** 10.3390/ijms21155385

**Published:** 2020-07-29

**Authors:** Christiaan M. Suttorp, René E. M. van Rheden, Natasja W. M. van Dijk, Maria P. A. C. Helmich, Anne Marie Kuijpers-Jagtman, Frank A. D. T. G. Wagener

**Affiliations:** 1Department of Dentistry—Orthodontics and Craniofacial Biology, Radboud University Medical Center, 6525 EX Nijmegen, The Netherlands; Maarten.Suttorp@radboudumc.nl (C.M.S.); Rene.vanRheden@radboudumc.nl (R.E.M.v.R.); Natasja.vanDijk@radboudumc.nl (N.W.M.v.D.); pia.helmich18@telfortglasvezel.nl (M.P.A.C.H.); 2Radboud Institute for Molecular Life Sciences, Radboud University Medical Center, 6525 GA Nijmegen, The Netherlands; 3Department of Orthodontics, University of Groningen and University Medical Center Groningen, 9713 GZ Groningen, The Netherlands; anne-marie.kuijpers@zmk.uni.be; 4Department of Orthodontics and Dentofacial Orthopedics, University of Bern, CH-3010 Bern, Switzerland; 5Faculty of Dentistry, Universitas Indonesia, Jakarta ID-10430, Indonesia

**Keywords:** embryology, cleft palate, abortion, placenta, vascular inflammation, heme, heme oxygenase, oxidative stress, inflammatory stress, ICAM-1, macrophage

## Abstract

Both infectious as non-infectious inflammation can cause placental dysfunction and pregnancy complications. During the first trimester of human gestation, when palatogenesis takes place, intrauterine hematoma and hemorrhage are common phenomena, causing the release of large amounts of heme, a well-known alarmin. We postulated that exposure of pregnant mice to heme during palatogenesis would initiate oxidative and inflammatory stress, leading to pathological pregnancy, increasing the incidence of palatal clefting and abortion. Both heme oxygenase isoforms (HO-1 and HO-2) break down heme, thereby generating anti-oxidative and -inflammatory products. HO may thus counteract these heme-induced injurious stresses. To test this hypothesis, we administered heme to pregnant CD1 outbred mice at Day E12 by intraperitoneal injection in increasing doses: 30, 75 or 150 μmol/kg body weight (30H, 75H or 150H) in the presence or absence of HO-activity inhibitor SnMP from Day E11. Exposure to heme resulted in a dose-dependent increase in abortion. At 75H half of the fetuses where resorbed, while at 150H all fetuses were aborted. HO-activity protected against heme-induced abortion since inhibition of HO-activity aggravated heme-induced detrimental effects. The fetuses surviving heme administration demonstrated normal palatal fusion. Immunostainings at Day E16 demonstrated higher numbers of ICAM-1 positive blood vessels, macrophages and HO-1 positive cells in placenta after administration of 75H or SnMP + 30H. Summarizing, heme acts as an endogenous “alarmin” during pregnancy in a dose-dependent fashion, while HO-activity protects against heme-induced placental vascular inflammation and abortion.

## 1. Introduction

Cleft lip and palate (CLP) is a serious orofacial birth defect affecting approximately 1/750 newborns worldwide with ethnic and geographical variation [1]. CLP is associated with various problems related to facial growth, feeding, speech, hearing and psychosocial status [2]. Palatogenesis is a dynamic and complex process and requires elevation, growth and adhesion of both palatal shelves, formation of the midline epithelial seam (MES), disintegration of this MES and arrangement of mesenchymal confluence. Failure of one of these events results in palatal clefting [3,4]. Both genetic and environmental factors play a role in the etiology of CLP [5]. Environmental factors such as infections, maternal smoking and alcohol use may increase the risk of congenital abnormalities, including CLP [6,7]. Both infectious and non-infectious placental inflammation can cause pregnancy complications and poor perinatal outcome [8].

During the first trimester of human gestation, intrauterine hematoma is commonly present [9,10], causing the release of large amounts of the alarmin heme [11,12]. We and others previously demonstrated that this free heme is responsible for a wide variety of inflammatory pathologies, including adverse effects in sickle cell disease, malaria, sepsis and kidney failure [13,14,15,16,17]. Heme was revealed to have the potential of threatening vascular endothelial cells [18], and oxidation of hemoglobin to methemoglobin was shown to lead to the liberation of heme moieties [19], triggering a pro-oxidant and cytotoxic environment in the vasculature [20,21]. Oxidation of heme to ferryl state also provokes inflammatory response [22,23]. Free heme administration was found to generate vascular oxidative and inflammatory stress following TLR4 activation, resulting in hampered wound repair, intrauterine fetal growth restriction and fetal abortion [11,24,25,26,27]. Intrauterine hematoma/hemorrhage is associated with an increased risk of early and late pregnancy loss [28], although its volume, location, gestational age, duration and the presence of vaginal bleeding determine the pregnancy outcome [9,29,30].

To protect against the adverse effects of free heme diverse defense systems are present. Free hemoglobin and heme molecules are first scavenged by haptoglobin (Hp) and hemopexin (Hpx), respectively [31,32], whereas the Breast Cancer Resistance Protein (BCRP) may further protect against heme-induced toxicity by exporting heme molecules across the plasma membrane [17]. The injurious effects of free heme molecules may also be counteracted by the antioxidant enzyme system heme oxygenase (HO) [33]. HO is a key regulator during embryonic development including placentation, spiral artery remodeling and blood pressure control [25,34,35]. Reduced HO-activity in the human placenta is associated with placental inflammation, including expression of adhesion molecules, influx of inflammatory cells, pre-eclampsia, fetal growth retardation and even abortion [34,36,37,38]. HO-1 is the highly inducible isoform, while HO-2 is constitutively expressed [39]. We previously demonstrated that abrogation of HO-2 expression in mice led to fetal growth restriction and craniofacial anomalies [40].

By degrading heme, the HO system generates the products free iron/ferritin, carbon monoxide, and biliverdin/bilirubin [32,34]. These HO effector molecules upregulate vasodilation and anti-apoptotic signaling, inhibit platelet aggregation and reduce leukocyte adhesion and pro-inflammatory cytokine expression [38]. Heme release following kidney surgery in mice was found to activate the inflammasome in macrophages by inducing the cytokine IL-1β [41], while HO-1 inhibited the induction of IL-1β in acute inflammatory arthritis in mice [42].

We postulate that exposure of pregnant mice to heme during palatogenesis initiates oxidative and inflammatory stress, leading to pathological pregnancy, increasing the incidence of palatal clefting and abortion. We expect that HO-activity protects against heme-induced pregnancy pathology, while HO-activity inhibitors exacerbate heme-induced inflammatory insults.

## 2. Results

### 2.1. HO-Activity Protects against Heme-Induced Abortion

We studied the consequences of HO-activity inhibition and the dose-dependent effects of heme administration on fetal development in mice. Higher fetal loss rate was found in the 75H group (*p* < 0.01) compared to the control group where just one fetal resorption in total was found (see Figure 1). In both the 150H group (*p* < 0.001) and SnMP + 75H group (*p* < 0.001), only abortions were observed, which was significantly more compared to the 75H group. In mice receiving SnMP, no abortions were found.

The observed total abortions corresponded with the body weight of the plugged mice between Days E12 and E15 (see Figure 2A). As expected, compared to the non-pregnant mice, the pregnant mice showed an increase in body weight (*p* < 0.001), whereas mice that demonstrated total abortion showed body weight reduction (*p* < 0.01). No adult mice died after heme and/or SnMP administration. The treated plugged mice demonstrated normal behavior, and the different experimental groups scored only a low to mild degree of discomfort.

Fewer fetuses per pregnant mouse in the 75H group (*p* < 0.01) were obtained compared to the controls (see Figure 2B). In the other experimental groups (30H, SnMP and SnMP + 30H), the fetal number was also lower compared to the controls, but this trend did not reach statistical significance (*p* > 0.05).

### 2.2. Placental Weight Increase after HO-Activity Inhibition

Considering that downregulation of HO-1 and HO-2 expression in humans’ placenta is associated with placental pathology [34,36,37,38], and placentas from HO-1^+/−^ mice were lighter than those from wt mice [43], we studied the effect of heme administration and the absence or presence of HO-activity at the mouse placental weight. The placentas were heavier in the SnMP group (*p* < 0.001) compared to the groups without SnMP administration, i.e., control, 30H and 75H groups (see Figure 2C). A higher placental weight was found in the SnMP + 30H group (*p* < 0.01) compared to the control and 30H groups.

### 2.3. Fetal Body Size Increase after Inhibition of HO-Activity and Heme Administration

Because downregulation of HO-1 expression in human placentas is associated with pregnancy complications and mouse fetuses from HO-1^+/−^ pregnancies were smaller than those from wt pairings [43], the effect of HO-activity inhibition and/or heme administration on mouse fetal body size was examined. Administration of SnMP increased the fetal body weight (*p* < 0.001) and fetal body length (*p* < 0.05) compared to the controls (see Figure 2D,E, respectively). Administration of 75H also significantly increased fetal body weight (*p* < 0.001) and fetal body length (*p* < 0.001) compared to the controls. A representative fetus per group is shown in Figure 3A.

### 2.4. Palatal Fusion despite Inhibition of HO-Activity and Heme Administration

As activation of pro-inflammatory pathways increased the risk of palatal clefting in humans [6,7,44], the effects of inhibition of HO-activity and/or heme administration on palatal fusion was studied in mice. The palatal shelves were fused in the fetuses at Day E16, and disintegration of the MES was found in palatal sections from fetuses of both the control and the different experimental groups: 30H, 75H, SnMP and SnMP + 30H. A representative HE stained palatal section per group is shown in Figure 3B. In the surviving fetuses, no effects on palatal fusion was found despite inhibition of HO-activity and/or exposure to heme.

### 2.5. Inhibition of HO-Activity Reduces IL-1β Expression in Placental Blood Vessels

Others found that IL-1β induction is counteracted by HO-1 activity during acute inflammatory arthritis in mice [42]. Here, the effects of exposure to SnMP and/or different heme doses, respectively, five and four days after administration, on IL-1β expression in mouse placenta were studied. Lower numbers of IL-1β positive blood vessels were found after administration of SnMP (median = 2.91/mm^2^) compared to the control (median = 11.0/mm^2^; *p* < 0.01) and 30H group (median = 13.1/mm^2^; *p* < 0.001) (see Figure 4A,B). Lower numbers of IL-1β positive blood vessels were also found after administration of SnMP + 30H (median = 4.13/mm^2^) compared to the 30H group (median = 13.1/mm^2^; *p* < 0.01).

### 2.6. Heme-Induced ICAM-1 Expression in Placental Blood Vessels Is Counteracted by HO-Activity

Since we and others previously demonstrated that heme can induce adhesion molecule expression in vascular endothelial cells [26,45,46], we examined whether heme exposure and/or HO-activity inhibition could alter the expression of ICAM-1 in placenta in mice. Higher numbers of ICAM-1 positive blood vessels were found after administration of 75H (mean= 6.6/mm^2^; *p* < 0.05) compared to the controls (see Figure 4A,C). In addition, more ICAM-1 positive blood vessels were found after administration of SnMP + 30H (mean = 10.3/mm^2^; *p* < 0.001) compared to the control (mean = 1.43/mm^2^), 30H (mean = 2.76/mm^2^) and SnMP groups (mean = 2.87/mm^2^).

### 2.7. Placental Macrophage Recruitment Increases after Heme Administration and Decreased HO-Activity

Since ICAM-1 expression was increased following exposure to 75H and SnMP + 30H, we assessed whether there was also enhanced influx of macrophages in the activated endothelium. Downregulation of HO-activity in human placenta is associated with inflammatory cell influx [34,36]. We therefore studied the effects of HO-activity inhibition and heme administration at the placental macrophage influx in mice. Higher numbers of macrophages were found after administration of 75H (mean = 4.6/mm^2^) compared to the control (*p* < 0.05), 30H (*p* < 0.001) and SnMP groups (*p* < 0.001) (see Figure 4A,D). Increased macrophage influx was also found after administration of SnMP + 30H (mean = 5.3/mm^2^; *p* < 0.001) compared to the control (mean = 1.6/mm^2^), 30H (mean = 0.8/mm^2^) and SnMP groups (mean = 0.6/mm^2^).

### 2.8. Placental HO-1 Expression Increases after Heme Administration

Since heme, the substrate of the HO system, was found to induce HO-1 expression [32,34,38], we investigated whether heme administration resulted in upregulation of placental HO-1 expression. Higher numbers of HO-1 positive cells were found after administration of 75H (median = 10.0/mm^2^) compared to the control (median = 0.7/mm^2^; *p* < 0.001) and 30H group (median = 1.5/mm^2^; *p* < 0.05) (see Figure 4A,E). Higher numbers of HO-1 positive cells were also found after administration of SnMP + 30H (median = 7.4/mm^2^) compared to the control (*p* < 0.001) and 30H group (*p* < 0.05). Notably, HO-1 positive cells were predominantly found in the perimetrium, the thin outer epithelial cell layer of the placenta.

## 3. Discussion

This study demonstrated that mimicking intrauterine hemorrhage/hematoma by i.p. administration of the endogenous “alarmin” heme, at Day E12 in pregnant mice, causes abortion/resorption in a dose-dependent fashion. No effect on the fusion of the palatal shelves was found in the surviving fetuses. Since we found a fetal loss rate of 50% in the 75H group, we were only able to study palatogenesis at Day E16 in the limited number of surviving fetuses.

Harmful effects of heme exposure have also been found in humans since uterine hematomas are associated with increased risk at intrauterine growth restriction, preterm delivery and miscarriage [47,48,49,50]. In our study, HO-activity rescued fetuses from heme-induced abortion. Administration of 75H led to fetal loss in half of the fetuses, however, blocking the HO activity by SnMP prior to 75H administration resulted in total abortion. Blocking of HO-activity by i.p. injection of zinc mesoporhyrin in mice earlier during pregnancy at Days E0 and E3 [51], E4 and E6 was previously found to increase the abortion rate [36,51]. In mice undergoing abortion, downregulation of both HO-1 and HO-2 was found in placental tissue compared to normal pregnant mice at Day E14 [52]. We previously found fetal growth restriction, severe malformations and craniofacial anomalies in HO-2 KO fetuses at Day E15 [40]. By contrast, upregulation of HO-1 by cobalt-protoporphyrin [36], or exposure to carbon monoxide [53] a product of heme break-down, rescued abortion-prone CBA/J × DBA/2J fetuses from abortion.

Notably, in our study no increased abortion was found after i.p. injection of SnMP at E11 alone. By contrast, the fetal body size was even increased after HO-activity inhibition, suggesting that HO-activity might be more crucial during implantation compared to fetal development at a later stage. Interestingly, others showed that adult HO-2 KO mice were obese, induced by disrupted metabolic homeostasis, caused by insulin resistance and elevated blood pressure [54]. Furthermore, in HO-2 KO mice increased brain edema was observed after intracerebral hemorrhage [55]. In rats resuscitated from cardiac arrest, induction of HO-1 by hemin reduced brain edema, improved neurologic outcome [56]. In contrast, SnMP administration reduced intracerebral mass in an intracerebral hemorrhage model in pigs by decreasing both hematoma and edema volumes [57]. Although the literature shows conflicting results, we cannot rule out that edema formation contributed to the fetal body size increase after HO-activity inhibition. However, we show here that HO-activity is also crucial for protecting against the injurious actions of heme at later stages during embryonic development.

Similarly, i.p. injection of another TLR4-ligand, lipopolysaccharide (LPS) increased abortion in rats and mice in a dose-dependent matter [58,59]. However, administration of different anti-inflammatory drugs protected against LPS-induced abortion. In rats, administration of azithromycin [60] and, in mice, administration of vitamin D3 [61], berberine [62], Pre-Implantation Factor (PIF) [63], curcumin [64], sildenafil [65] or heparin [65] protected from LPS-induced abortion. For example, HO-1 expression in mouse placenta demonstrated also a dose- and time-dependent protection following exposure to LPS [66]. Interestingly, almost all of these protecting agents are also potent HO-1 inducers [67,68,69,70,71], suggesting that HO-1 also protects against the injurious effects of other TLR4-ligands besides heme.

In human placental endothelial cells, exposure to LPS was found to induce IL-1β expression within 24 h in vitro [72,73]. Increased levels of IL-1β were found in placental endothelial cells following exposure for 24 h to trophoblast debris from preeclamptic placentae in vitro [74]. In human pregnancies with reduced fetal movement increased placental expression of IL-1β was observed compared to the controls, although some controls also demonstrated some IL-1β expression [75].

Expression of the pro-inflammatory cytokine IL-1β was observed in blood vessels in the placentas of the controls, as well as in placentas exposed to 30H and 75H. Recent data show that heme exposure triggers the NLRP3 inflammasome pathway, inducing IL-1β production in human endothelial cells in vitro [76]. Intrauterine infections are able to trigger the expression of IL-1β, which stimulates uterine contractions [77], by elevating PGE_2_ levels and myometrial contractility [78]. By contrast, HO-1 was found to inhibit IL-1β induction during acute inflammatory arthritis in mice [42], however, we observed that HO-1 inhibition by administration of SnMP decreased placental IL-1β expression in mice. Unfortunately, a limitation of our study was that the effects of SnMP and heme administration were only evaluated five and four days, respectively, after administration. This means that we were not able to evaluate the expression of IL-1β shortly after SnMP and/or heme administration, with the risk that this expression has been dampened.

Both administration of 75H and SnMP + 30H increased placental vascular ICAM-1 expression and macrophage influx. However, no increases in ICAM-1 and macrophage influx were found after administration of heme 30, showing that heme-induced ICAM-1 expression is dose-dependent and likely related to scavenging by Hpx or HO-mediated breakdown at lower concentrations. When ICAM-1 expression together with the number of macrophages in placenta was elevated in sound-stressed mice, increased abortion was found [79]. Moreover, decreased abortion was observed following i.p. injection of monoclonal antibodies to ICAM-1 in mice, indicating that ICAM-1 facilitates immunologically-mediated abortion [80]. Increased levels of ICAM-1 were found in placental endothelial cells following exposure to trophoblast debris from preeclamptic placentae in vitro [74]. In women suffering from Morbidly Adherent Placenta (MAP), an abnormal adherence of the placenta to the myometrium, massive hemorrhage can occur, resulting in severe morbidity and mortality and increased ICAM-1 expression [81]. In addition, in serum [82,83,84,85] and placenta [85,86] from preeclampsia patients, increased ICAM-1 expression was found.

Placental inflammation in uncomplicated human gestations was only present in 4% of the cases [87]. Intrauterine infection induces pro-inflammatory cytokines, particularly IL-1β and TNFα, and influx of macrophages in placenta and uterus, and it may lead to premature birth [77]. In mice, initiated preterm labor by intrauterine infusion of LPS provoked a massive influx of neutrophils in the myometrium [88]. Macrophage influx in placenta and myometrium is associated with placental inflammation and preterm labor in human, mouse and rat [89,90,91].

Our data show that HO-1 expression in placenta increased after administration of both 75H and SnMP + 30H. HO-1 positive cells were predominantly found in the perimetrium [92], indicating that besides the vasculature free heme molecules were able to reach the developing fetus through diffusion. In HO-1^+^/^−^ mice induction of HO-activity in placenta by administration of Pravastatin was shown to improve placental function and fetal survival at E14.5 [93]. In human pathological pregnancies, low expression of HO-2 stimulated migration of inflammatory cells to the feto-maternal interface, possibly caused by enhanced serum levels of free heme [34]. We speculate that in our experiment low levels of free heme were scavenged by Hpx [31], while the heme–Hpx complex binding to LRP1 would activate cytoprotective signaling by Nrf2 [94] and upregulation of placental HO-1 expression [32,95].

It is likely that in our experiment the sudden exposure to high doses of free heme overwhelmed the scavenger Hpx and the HO system [96], allowing binding to toll-like receptor 4 (TLR4) [46] and activation of nuclear factor kappa B (NF-κB) [97], driving expression of pro-inflammatory cytokines, such as IL-1β [98,99,100].

The novel hypothetical model (Figure 5) shows heme-induced endothelial cell activation [101], as exemplified by the expression of ICAM-1 [102], facilitating the recruitment of macrophages into placental tissue [102,103], resulting in pathological pregnancy, increasing the risk of abortion. Furthermore, heme-induced inflammation might cause craniofacial abnormalities before resorption of the fetuses would occur. However, our studies could not provide evidence for this statement. Inhibition of HO-activity will increase the heme-induced inflammatory response, albeit HO-activity will attenuate heme-induced inflammation, promoting normal fetal development.

## 4. Materials and Methods

### 4.1. Mice Selection, Mating, Housing, Ethical Permission

To obtain fetuses for this study, plugged CD1 outbred mice, 12–17 weeks of age, were purchased from Envigo, Venray, The Netherlands (*n* = 31), and from Charles River, Sulzfeld, Germany (*n* = 7). At the facility of the animal supplier, the female mice were mated with CD1 outbred male mice for 1 day and checked for the presence of a vaginal copulation plug on the next morning, taken as Day 0 of pregnancy (Gestational/Embryonic Day 0, E0) [104]. The plugged animals were transported to our animal facility and housed under specific pathogen-free housing conditions with 12 h light/dark cycle and ad libitum access to water and powdered rodent chow (Sniff, Soest, The Netherlands). The animals could acclimatize for at least 1 week before the start of the experiment. All mice were randomly assigned to the control or experimental groups by drawing lots. Ethical permission for the study was obtained according to the guidelines of the Board for Animal Experiments of the Radboud University Nijmegen (Ethical permission # RU-DEC 2012-166; date: 20 August 2012).

### 4.2. Heme and/or SnMP Administration, Sample Size, Animal Welfare Monitoring

To be able to study palatal clefting, fetuses should be studied beyond the time point palatal fusion takes place. In wt mice the palatal shelves fuse between embryonic Day E14.5 and E15.5, and the capacity to fuse is lost after Day E16 [105]. Fetuses of Day E16 were therefore considered to be suitable for studying palatal clefting.

Plugged CD1 females were subdivided for the heme administration in different doses: 30, 75 or 150 μmol/kg body weight (referred to as heme 30H, 75H and 150H, respectively). Heme was administered via intraperitoneal (i.p.) injection at Day E12 (see Figure 6).

Tin mesoporphyrin (SnMP) can bind strongly to the HO-1 and HO-2 enzyme, but cannot be broken down by both isotypes, therefore acting as a competitive inhibitor of HO-activity [106,107]. To abrogate the HO system prior to heme administration half of the animals of the 30H and 75H groups received SnMP 30 μmol/kg body weight (later referred to as SnMP) via i.p. injection at E11. In addition, the control group received neither heme nor SnMP, while another group received SnMP only. Heme and SnMP were purchased from Frontier Scientific, Carnforth, UK.

Heme and SnMP were freshly dissolved with Trizma base. The pH was adjusted to pH 7.6–8.0 with HCl. The heme and SnMP solution were filter sterilized before administration.

To detect an effect size of 0.25 (generalized estimation of the reduction in fetal and placental weight in the experimental groups) and a significance level of 0.05 for the 7 groups with power of 0.80, a total sample size of *n* = 231 fetuses for this study was calculated by the one-way ANOVA power analysis a priori (G*Power 3.1 software) [108]. This indicated that the mean sample size per group should comprise approximately 33 fetuses. We estimated that the litter size could range from 14 to 18 pups, suggesting that each group should contain at least 3 pregnant mice. The chance of conception was assumed to be around 70%, indicating that for each group at least 5 mice should be mated.

Animal welfare was monitored daily during the stay at our animal facility, and the degree of discomfort after administration was scored according to the guidelines of the Board for Animal Experiments of the Radboud University Nijmegen.

### 4.3. Adult Mouse Body Weight Comparison

At embryonic Days E12 and E15, the plugged mice were weighed. The body weight comparison, the percentage of the weight change between Days E12 and E15, of the pregnant mice, not pregnant mice and the mice that demonstrated total abortion was calculated. The status pregnant, not pregnant or total abortion was confirmed by examination of the uterus.

### 4.4. Fetal Loss Rate and Fetal Number Calculation

At Day E16, the animals were killed by CO_2_/O_2_ inhalation for 10 min. All fetuses were isolated from the uterus (see Figure 7A). The fetuses were separated from the placentas. For the fetus-carrying mice, the fetal loss rate, the percentage of non-viable and hemorrhagic embryonic implantations to the total number of embryonic implantations (non-viable or hemorrhagic embryonic implantations + fetuses), and the fetal number were calculated.

### 4.5. Placental Weight and Fetal Body Weight and Length

All fetuses and placentas were weighed. The body length of the fetuses was measured on photographs using Fiji Image J 1.51n software (National Institutes of Health, Bethesda, MD, USA) (see Figure 7B).

### 4.6. Paraffin Embedding and Section Cutting of Head and Placenta Samples

The fetuses were decapitated, the head samples and placentas were fixed for 24 h in 4% paraformaldehyde and further processed for routine paraffin embedding. Serial coronal sections of 5-µm thickness, through the secondary palate in head samples and through the middle part of the placenta, were mounted on Superfrost Plus slides (Menzel-Gläser, Braunschweig, Germany).

### 4.7. Hematoxylin-Eosin Staining of Palatal Sections and Immunohistochemical Stainings of Placental Sections

Selected paraffin embedded palatal and placental sections were deparaffinized using Histosafe (Adamas Instrumenten B.V., Rhenen, The Netherlands) and rehydrated using ethanol (100-96%). Then, the endogenous peroxidase quenching step with 3% hydrogenperoxide in methanol was performed, followed by further rehydration by ethanol 96% till PBS. The palatal sections were routinely stained with Hematoxylin and Eosin (HE). Antigens were retrieved with citrate buffer at 70 °C for 10 min, followed by incubation in 0.015% trypsin in PBS at 35 °C for 5 min. Next, the placental sections were pre-incubated with 10% normal donkey serum (NDS) in phosphate-buffered saline with glycine (PBSG). Primary antibodies (Table 1) for cytokine IL-1β, vascular adhesion molecule-1 (ICAM-1), macrophage marker F4/80 [109] and HO-1 were diluted in 2% NDS in PBSG and incubated overnight at 4 °C. After washing with PBSG, sections were incubated for 60 min with a biotin-labeled secondary antibody (Table 2), as previously described [110]. Next, the sections were washed with PBSG and treated with avidin-biotin peroxidase complex (ABC) for 45 min in the dark. After extensive washing with PBSG, diaminobenzidine (DAB)-peroxidase staining was performed for 10 min. Finally, the nuclei were counterstained with Hematoxylin for 10 s and sections were rinsed for 10 min in water, dehydrated and embedded in distyrene plasticizer xylene (DPX).

Microscopic photographs of the HE and immunohistochemical stained sections were taken using a Carl Zeiss Imager Z.1 system (Carl Zeiss Microimaging GmbH, Jena, Germany) with AxioVision 4.8 v software (Zeiss, Göttingen, Germany).

The palatal sections were classified into 4 stages of palatogenesis based on the anatomy of the palatal shelves, as previously described by Dudas et al. [105]. Per individual fetus, the presence of palatal fusion was studied on multiple transversal sections.

### 4.8. Quantification of IL-1β, ICAM-1, F4/80 and HO-1 Immunoreactivity in Placental Sections

Because the transversal sections through the middle of the placenta showed a significant variance in size, immunoreactivity was adjusted to surface area. The specific surface area per placental section was measured using Fiji Image J 1.51n software (see Figure 7C). The number of IL-1β and ICAM-1 positive blood vessels and the number of F4/80 (macrophages) and HO-1 positive cells within the outline of the placental sections were counted, and the mean number per mm^2^ per group was calculated. The counting was performed twice, by two observers (C.M.S. and F.A.D.T.G.W.), independently and blinded for the groups. The inter- and intra-examiner reliability was determined.

### 4.9. Statistical Analysis

The data for fetal loss rate, adult mouse body weight comparison, the number of ICAM-1 positive blood vessels/mm^2^ and macrophages/mm^2^ in placental sections were quantitatively scored and showed a normal distribution as evaluated by the Kolmogorov–Smirnov test (KS-test). The data were analyzed using ANOVA and Tukey’s multiple comparison post hoc test to compare the differences between the groups.

The data for fetal number, placenta weight, fetal body weight, fetal body length, the number of IL-1β positive blood vessels/mm^2^ and HO-1 positive cells/mm^2^ in placental sections were quantitatively scored and showed a non-normal distribution as evaluated by the KS-test. The data were analyzed using the non-parametric Kruskal–Wallis ANOVA on ranks test and Dunn’s Multiple Comparison post hoc test to compare differences between the groups.

To determine the inter- and intra-examiner reliability, the coefficient of determination (*R*^2^) was calculated by the square of the Pearson correlation coefficient, and acceptable scores > 0.80 were obtained for the counting.

Differences were considered to be significant if *p* < 0.05. All statistical analyses of the data were performed using GraphPad Prism 5.03 software (GraphPad Software Inc.©, San Diego, CA, USA).

## 5. Conclusions

Our results show that heme acts as an endogenous “alarmin” during pregnancy in a dose-dependent fashion, while HO-activity protects against heme-induced placental vascular inflammation and abortion in mice. More research is necessary to unravel the precise relation between hemorrhage-induced placental inflammation, pathological pregnancy and palatal clefting.

## Figures and Tables

**Figure 1 ijms-21-05385-f001:**
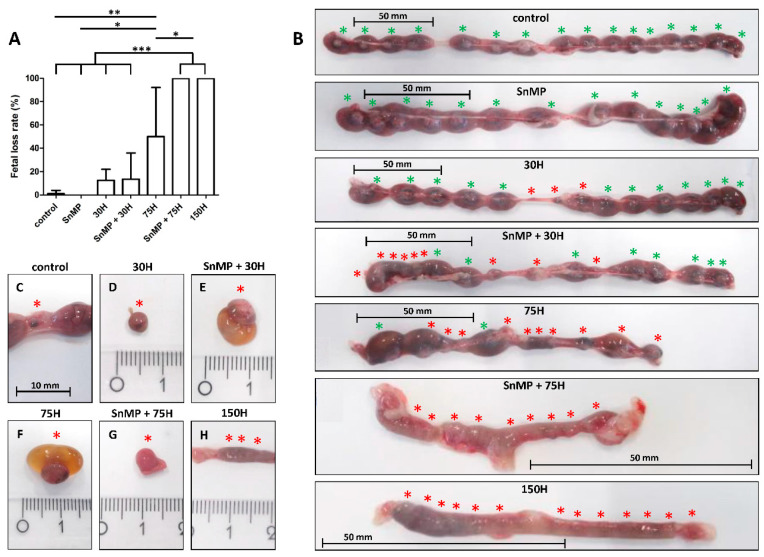
HO-activity protects against heme-induced abortion. (**A**) Bar chart of the fetal loss rate in pregnant mice determined at E16, compared for the different heme and/or SnMP administrations (control *n* = 6; SnMP *n* = 4; 30H *n* = 4; SnMP + 30H *n* = 5; 75H *n* = 7; SnMP + 75H *n* = 4; 150H *n* = 3). Data are shown as mean ± SD. * *p* < 0.05, ** *p* < 0.01, *** *p* < 0.001. (**B**) The isolated uteri were photographed, the green asterisk indicated a fetus, the red asterisk indicated a fetal resorption. A representative uterus per group is shown. (**C**–**G**) Isolated fetal resorptions from different groups (centimeter ruler). Fetal resorption showing an amniotic sac that contained an abnormal yellow-brownish pigmentation of the amniotic fluid (**E**,**F**), characteristic for the presence of free heme molecules. (**H**) Part of the uterus containing an amorphic mass as a result of total abortion (centimeter ruler).

**Figure 2 ijms-21-05385-f002:**
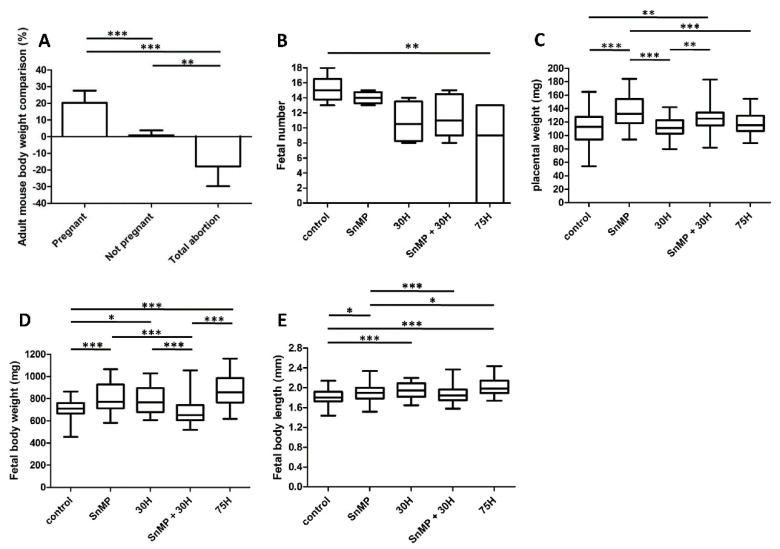
Adult mouse body weight comparison, fetal number, fetal body weight and placental weight affected by HO-activity inhibition and heme administration. (**A**) Bar chart for the adult mouse body weight comparison, the percentage (%) of the weight change between Days E12 and E15. Unfortunately, the weight at Day E15 was unintentionally not collected from two pregnant mice (75H group) and two mice that demonstrated abortion (SnMP + 75H and 75H groups). Eventually, the body weight comparison of the pregnant mice (*n* = 22; control *n* = 6, SnMP *n* = 4, 30H *n* = 4, SnMP + 30H *n* = 5, 75H *n* = 3), not pregnant mice (*n* = 5; SnMP = 1, 30H *n* = 2, SnMP + 30H *n* = 2) and mice that demonstrated total abortion (*n* = 7; 75H *n* = 1, SnMP + 75H *n* = 3, 150H *n* = 3) was calculated. Data are shown as mean ± SD. (**B**) Box-and-whisker plot with 10–90 percentiles of quantitative assessment of the fetal number (number of fetuses per pregnant mouse of the different groups). Number of pregnant mice per group; control *n* = 6, SnMP *n* = 4, 30H *n* = 4, SnMP + 30H *n* = 5, 75H *n* = 7. Number of fetuses per group: control *n* = 91, SnMP *n* = 56, 30H *n* = 43, SnMP + 30H *n* = 58, 75H *n* = 46. (**C**) Box-and-whisker plot with 10–90 percentiles of quantitative assessment of the placental weight of the different groups (control *n* = 91, SnMP *n* = 56, 30H *n* = 43, SnMP + 30H *n* = 58, 75H *n* = 46). Box-and-whisker plot with 10–90 percentiles of quantitative assessment of the (**D**) fetal body weight and (**E**) fetal body length of the fetuses of the different groups (control *n* = 91, SnMP *n* = 56, 30H *n* = 43, SnMP + 30H *n* = 58, 75H *n* = 46) were measured. * *p* < 0.05, ** *p* < 0.01, *** *p* < 0.001.

**Figure 3 ijms-21-05385-f003:**
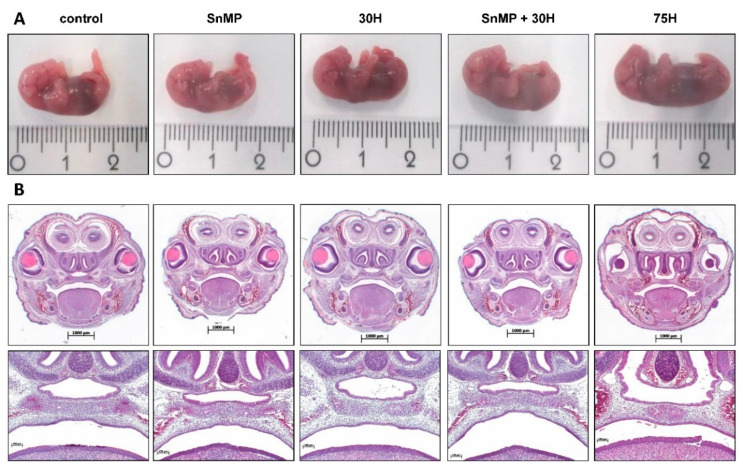
Palatal fusion despite inhibition of HO-activity and heme administration. (**A**) A representative fetus per group is shown: fetus control, 1.96 cm, 659.5 mg; fetus SnMP, 1.94 cm, 846.3 mg; fetus 30H, 1.95 cm, 832.3 mg; fetus SnMP + 30H, 1.69 cm, 654.1 mg; fetus 75H, 2.43 cm, 1125.9 mg (centimeter ruler). (**B**) The palatal sections were classified into four stages of palatogenesis based on the anatomy of the palatal shelves: elevation, horizontal growth, midline adhesion and fusion. Representative HE staining palatal sections (magnification: ×100) of the different groups: control; SnMP; 30H; SnMP + 30H; 75H. Fusion of the shelves of the secondary palate, together with disintegration of the MES, was observed in all fetuses of the control and different experimental groups. A minimum of five transversal sections from at least five head samples per group was assayed.

**Figure 4 ijms-21-05385-f004:**
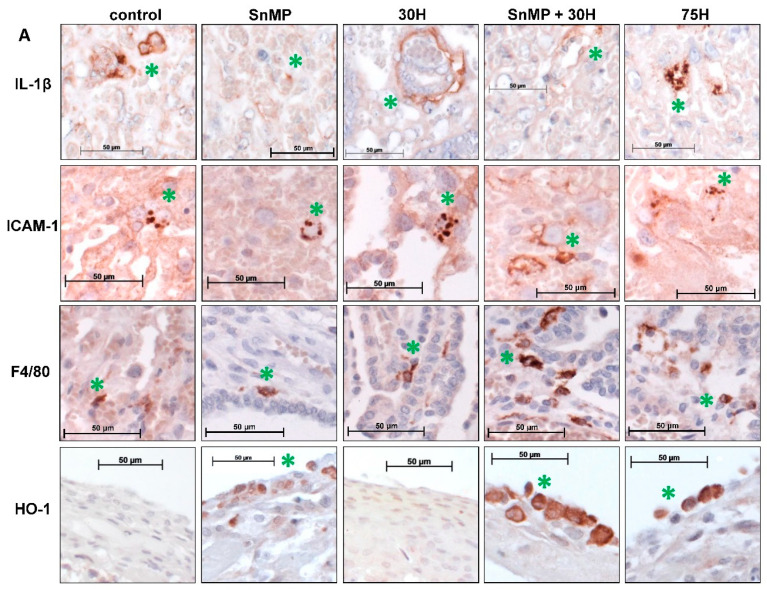
Quantification of IL-1β, ICAM-1, F4/80 and HO-1 immunoreactivity in placenta. (**A**) Placental sections stained for immunoreactivity of IL-1β, ICAM-1, F4/80 and HO-1. Representative IL-1β, ICAM-1, F4/80 and HO-1 immunostaining in placenta from the control and experimental groups. Green asterisk indicates IL-1β positive blood vessel, ICAM-1 positive blood vessel, F4/80 positive macrophage and HO-1 positive cell, respectively. (**B**) Box-and-whisker plot with 10–90 percentiles of quantitative assessment of the IL-1β positive blood vessels/mm^2^ compared for the different heme and/or SnMP administrations. (**C**) Bar chart of ICAM-1 positive blood vessels/mm^2^ compared for the different heme and/or SnMP administrations. Data are shown as mean ± SD. (**D**) Bar chart of number of macrophages/mm^2^ compared for the different heme and/or SnMP administrations (control *n* = 6; SnMP *n* = 7; 30H *n* = 6; SnMP + 30H *n* = 7; 75H *n* = 5). Data are shown as mean ± SD. (**E**) Box-and-whisker plot with 10–90 percentiles of quantitative assessment of the HO-1 positive cells/mm^2^ compared for the different heme and/or SnMP administrations. Quantification of IL-1β, ICAM-1, F4/80 and HO-1 immunoreactivity in placenta (Control *n* = 6; SnMP *n* = 7; 30H *n* = 6; SnMP + 30H *n* = 7; 75H *n* = 5). * *p* < 0.05, ** *p* < 0.01, *** *p* < 0.001. A minimum of five sections from a minimum five placentas per group was assayed.

**Figure 5 ijms-21-05385-f005:**
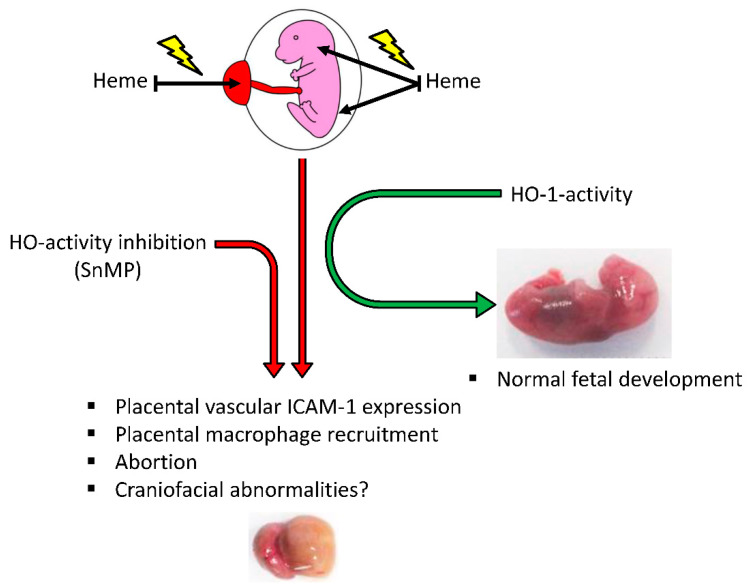
Hypothetical model: Heme oxygenase protects against placental vascular inflammation and abortion by the alarmin heme in mice. This study demonstrated that heme acted as an endogenous “alarmin” during pregnancy in a dose-dependent fashion, while HO-activity protected against heme-induced placental vascular inflammation and abortion. We postulate that heme-induced inflammatory response promoted endothelial cell activation, which upregulated the expression of ICAM-1, facilitating the recruitment of macrophages into placental tissue, resulting in a pathological pregnancy, increasing the risk of abortion and possibly craniofacial abnormalities. Inhibition of HO-activity will increase the heme-induced inflammatory response, albeit HO-activity will attenuate heme-induced inflammation, promoting normal fetal development.

**Figure 6 ijms-21-05385-f006:**
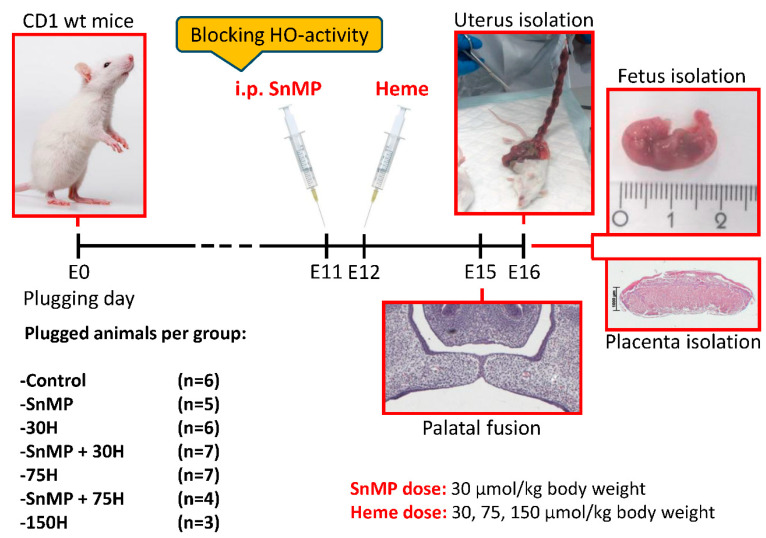
Administration of heme and/or SnMP in plugged mice via intraperitoneal injection. Plugged mice received SnMP (30 μmol/kg body weight) at Day E11 and/or heme in different dose (30, 75 and 150 μmol/kg body weight) at Day E12 by i.p. injection. All plugged mice were killed by CO_2_/O_2_ inhalation for 10 min at Day E16; the uteri were removed; and the fetuses and placentas were isolated. The first animals that received administration of SnMP + 75H (*n* = 4) or 150H (*n* = 3) demonstrated resorption of all fetuses. Since no fetuses could be obtained, the remaining animals of both groups (*n* = 4) were reassigned to the SnMP + 30H group and 75H group. The plugged animals were eventually divided into 7 groups as follows: control *n* = 6, SnMP *n* = 5, 30H *n* = 6, SnMP + 30H *n* = 7, 75H *n* = 7, SnMP + 75H *n* = 4, 150H *n* = 3. In total, 38 uteri were removed. In 1 mouse from the SnMP group, 2 mice from the 30H group and 2 mice from the SnMP + 30H group, no non-viable/hemorrhagic embryonic implantations or fetuses were found in the uterus, and those animals were regarded as not pregnant. Furthermore, in 2 mice of the 75H group, total abortion had occurred. Finally, from 24 mice (control *n* = 6, SnMP *n* = 4, 30H *n* = 4, SnMP + 30H *n* = 5, 75H *n* = 5) in total, 294 fetuses and placentas were obtained (control *n* = 91, SnMP *n* = 56, 30H *n* = 43, SnMP + 30H *n* = 58, 75H *n* = 46).

**Figure 7 ijms-21-05385-f007:**
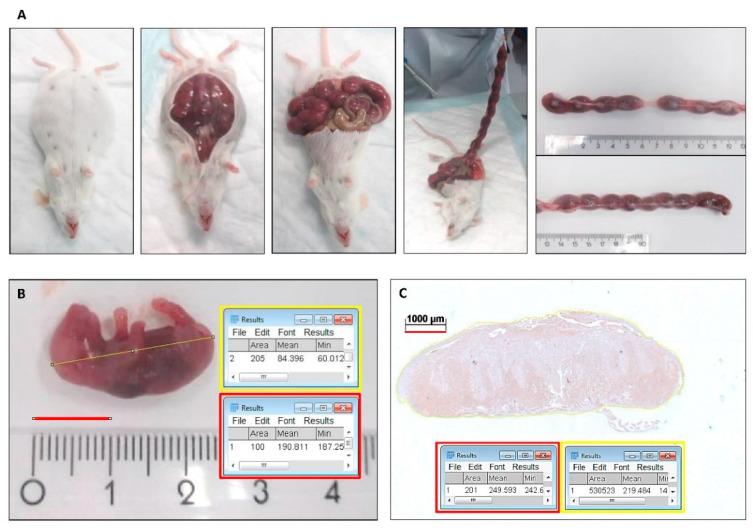
Isolation of fetuses and placentas and measurement of fetal body length and placental section surface area. (**A**) After sacrifice of the plugged mice at Day E16, the uteri were isolated (centimeter ruler). (**B**) The fetuses were separated from the placentas, photographed and a scale bar was drawn at the centimeter ruler of 10 mm, and the total number of pixels of the bar was determined (e.g., 30H; 100 pixels). A line depicting the length of the body of the fetus was drawn and the number of pixels was recorded (e.g., 205 pixels). The body length was calculated (e.g., 205/100 = 2.05 cm). (**C**) Placental sections surface area measurement. Coronal sections through the middle of the placenta (e.g., SnMP and ICAM-1 immunostaining). A scale bar was drawn at the ruler in each photograph of 1000 μm and the total number of pixels of the bar was determined (e.g., 201 pixels). The surface of 1000 µm × 1000 μm (1 mm^2^) included the total number of 40,401 (201 × 201) pixels. The outline of the total placenta was drawn, and the number of pixels was recorded (e.g., 530,523 pixels). Next, the total section surface was calculated (e.g., 530,523/40,401 = 13.1 mm^2^).

**Table 1 ijms-21-05385-t001:** Primary antibodies used for immunohistochemical staining for IL-1β, ICAM-1, F4/80 and HO-1. Source and used concentrations of the antibodies are described.

First Antibody	Specificity	Concentration (µg/mL)	Source
sc-1252	IL-1B	0.2	Santa Cruz Biotechnology, Santa Cruz, CA, USA
CD54	ICAM-1	0.34	Proteintech via Sanbio, Uden, The Netherlands
A3-1 (ab6640)	F4/80	1.0	Abcam Cambridge Biomedical Campus, Cambridge, UK
SPA 895	HO-1	1.0	Stressgen, Victoria, BC, USA

**Table 2 ijms-21-05385-t002:** Secondary antibodies used for immunohistochemical staining for IL-1β, ICAM-1, F4/80 and HO-1. Source and used concentrations of the antibodies are described.

Secondary Antibody	Specificity	Concentration (µg/mL)	Source
705-065-147	Donkey anti-goat Biotin	2.8	Jackson Immunoresearch West Grove, PA, USA
711-065-147	Donkey anti-rabbit Biotin	2.0	Jackson Immunoresearch West Grove, PA, USA
712-065-153	Donkey anti-rat Biotin	4.6	Jackson Immunoresearch West Grove, PA, USA
711-065-147	Donkey anti-rabbit Biotin	2.0	Jackson Immunoresearch West Grove, PA, USA

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
