# Peer review of "Heme Oxygenase Protects against Placental Vascular Inflammation and Abortion by the Alarmin Heme in Mice"

_ijms, 2020, doi:10.3390/ijms21155385_

Round 1

Reviewer 1 Report

The authors analyzed the protective role of heme oxygenase-1 against placental vascular inflammation and abortion triggered by heme. The methodology of the work is appropriate and allows the replication. The references appropriately cover the literature. The conclusions of the study are original and clear without flaws. The present work established an important direct link between heme oxygenase-1 and its protective effect in heme-induced placental vascular inflammation and abortion that is supported by a carefully selected set of experiments.

Suggested improvement:

Are the increases in fetal body weights caused by oedema?

Author Response

Dear reviewer 1,

In main document the possibility that oedema has caused the increase in fetal body weight is discussed (line 235-242), see attachment.

Reviewer 2 Report

The authors have done a good job of putting together a well-designed experiment and a nicely written manuscript. There are no major changes suggested. However, there are a couple of typos that could be corrected, page one, line 25 'proof' should be 'test', and page two, line 59 'defence' should be 'defense.' This is a solid paper that will contribute to the field.

Author Response

Dear Reviewer 2,

-Line 25, proof has been corrected into test.

-Line 59, defence has been corrected into defense

See also the changes in the main document, see attachment. 
